# CAN AGENT LEARN ROBUST LOCOMOTION SKILLS WITHOUT MODELING ENVIRONMENTAL OBSERVATION NOISE?

## ABSTRACT

Deep Reinforcement Learning (DRL) has been widely attempted for solving locomotion control problems recently. Under the circumstances, DRL agents observe environmental measurements via multi-sensor signals, which are usually accompanied by unpredictable noise or errors. Therefore, well-trained policies in simulation are prone to collapse in reality. Existing solutions typically model environmental noise explicitly and perform optimal state estimation based on this. However, there exists non-stationary noise which is intractable to be modeled in real-world tasks. Moreover, these extra noise modeling procedures often induce observable learning efficiency decreases. Since these multi-sensor observation signals are universally correlated in nature, we may use this correlation to recover optimal state estimation from environmental observation noise, and without modeling them explicitly. Inspired by *multi-sensory integration* mechanism in mammalian brain, a novel **S**elf-supervised random**I**zed **M**asked **A**ugmentation (**SIMA**) algorithm is proposed. SIMA adopts a self-supervised learning approach to discover the correlation of multivariate time series and reconstruct optimal state representation from disturbed observations latently with a theoretical guarantee. Empirical study reveals that SIMA performs robust locomotion skills under environmental observation noise, and outperforms state-of-the-art baselines by **15.7%** in learning performance.

## 1 INTRODUCTION

Owing to powerful function approximation capacity of deep neural networks, Deep Reinforcement Learning (DRL) has been demonstrated to achieve great success on many complex tasks, including board games Silver et al. (2016); Schrittwieser et al. (2020), electronic sports Vinyals et al. (2019); Berner et al. (2019), autonomous vehicle Fuchs et al. (2021); Wurman et al. (2022); Lillicrap (2016), and even air combat Sun et al. (2021); Pope et al. (2022). However, these remarkable achievements basically rely on a simulation environment that can provide correct state observations. In real-world tasks (e.g., robot locomotion control Hwangbo et al. (2019); Song et al. (2021a)), DRL agents observe environmental measurements via multi-sensor signals, which may contain uncertain noise that naturally originates from unpredictable sensor errors or instrument inaccuracy Zhang et al. (2020), e.g., considering a half-cheetah robot running in a field with multi-sensor signals as observations (illustrated in Figure 1). Once the robot landed after jumping, observation noise (i.e., stationary noise) might be involved in accelerometers due to collision and friction between its legs and ground. Besides, there also exists unpredictable disturbance (i.e., non-stationary noise) accidentally. Consequently, directly deploying well-trained DRL policy from simulation to reality might lead to catastrophic failures.

In view of this, a well-adopted approach is Domain Randomization (DR) Tobin et al. (2017); Andrychowicz et al. (2020); Song et al. (2021b); Tremblay et al. (2018). DR randomizes key parameters during simulation training that may change in the future, this randomization procedure provides extra robustness for the upcoming sim-to-real transfer. Another line of research is based on adversarial attack Kos & Song (2017); Huang et al. (2017); Mandlekar et al. (2017) to directly improve the robustness of DRL agents, e.g., Kos & Song (2017) first presents results of adversarial training on Atari environments using weak FGSM attacks on pixel space. These afore-

mentioned methods typically model environmental noise explicitly and perform optimal state estimation based on this. However, there exists non-stationary noise which is intractable to be modeled. Karra & Karim (2009). Therefore, prior works only work with stationary noise scenarios.

Considering the correlation among multi-sensor signal observations, one potential solution is to optimally estimate the disturbed signals from other clean signals by utilizing this internal-correlation mechanism, e.g., humans watch television by experiencing videos, sounds, and subtitles. Assuming that a literally error pops-up in current subtitle, human brain will correct the wrong subtitle via videos or sound playback latently, or even simply ignore the errors in subtitles. Accordingly, these errors from a single sensory channel merely affect the overall synthetic cognition process. This mechanism in *Superior Colliculus* neurons of mammalian brains is called *multi-sensory integration* Meredith & Stein (1983). *Multi-sensory integration* allows organisms to simultaneously sense and understand external stimuli from different modalities, which helps to reduce noise by combining information from different sensory modalities Stein et al. (2004); Koelewijn et al. (2010).

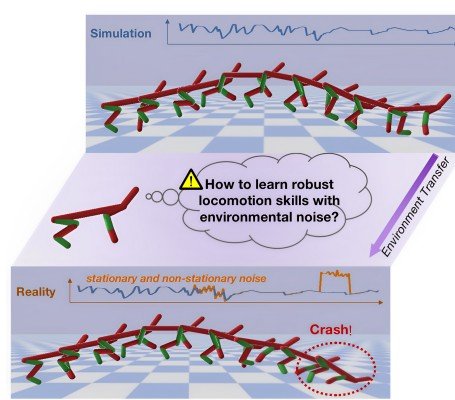

Figure 1: Due to environmental noise, a well-trained DRL-based half-cheetah robot lost its balance and crashed after jumping.

Inspired by *multi-sensory integration*, we focus on learning the internal-correlation of multivariate time series. This internal-correlation reconstructs the optimal state estimation representation latently from environmental noise via self-supervised learning. Essentially, this paper provides a paradigm shift that eliminates the requirement for explicit noise modeling and improves the robustness of DRL policy. Specifically, our contributions are summarized as three-fold:

- We first formulate the Markov-Decision Process (MDP) with environmental de-noising process as a **D**e-noising const**R**ained **P**olicy **O**ptimization **P**roblem (**DRPOP**), thereby, DRPOP converts this conventional state estimation into an optimal correlation representation learning form.
- Inspired by *multi-sensory integration* mechanism, a novel **S**elf-supervised random**I**zed **M**asked **A**rgumentation (**SIMA**) algorithm is proposed. SIMA learns internal-correlation of multivariate time series and reconstructs latent state representation from noisy observations. We also provide the reconstruction error is bounded by a variational evidence lower bound.
- Empirical study reveals that SIMA performs robust locomotion skills under environmental observation noise, and outperforms state-of-the-art baselines by **15.7**% in learning performance.

## 2 RELATED WORKS

### 2.1 DRL IN REAL-WORLD TASKS

A variety of DRL-based locomotion control methods have demonstrated promising performance in real-world applications, e.g., Lee et al. (2020); Hwangbo et al. (2019); Kumar et al. (2021); Miki et al. (2022) develop an automatic terrain curriculum combined with DRL algorithms, which enables quadruped robots to successfully traverse around complex natural terrains. There are also notable breakthroughs that exclusively involve training on physical robots Van Hoof et al. (2015); Falco et al. (2018); Jinxin Liu (2022). Nonetheless, due to the inherent slowness and costliness of physical trials, the acquired behaviors tend to exhibit substantial limitations. Xie et al. (2020); Haarnoja et al. (2018a) use DRL to simplify the design of locomotion controllers, automate parts of the design process, and learn behaviors that could not be engineered with prior approaches. Loquercio et al. (2019); Song et al. (2021a) demonstrate zero-shot sim-to-real transfer on the task of following a track with possibly moving gates at high speed. The abundance of simulated data, generated via DR, makes this system robust to changes of illumination and gate appearance. Akkaya et al. (2019); Andrychowicz et al. (2020) utilize DR to increase environmental diversity, including randomizing a series of environmental parameters. In conclusion, DRL policy can drive a shadow dexterous

hand to manipulate a cube dexterously. While the aforementioned works represent notable DRL adaptations for particular tasks, the present paper aims to provide a solution to improve system robustness against environmental noise.

## 2.2 State observation de-noising

Every stage of DRL may introduce uncertainty posing significant challenges to system robustness. This paper focuses on solving issues induced by environmental observation noise. To our best knowledge, three primary methods have been proposed for addressing this problem. Firstly, conventional methods employ filters tailored to the noise pattern in sensor measurements to eliminate observation noise Kalman (1960); Selesnick & Burrus (1998), or design disturbance observers Chen et al. (2015) to compensate for the input of locomotion controllers. These approaches rely on human expert's experience and accurate system dynamics model, which are usually difficult to obtain. Secondly, DR Tremblay et al. (2018); Tobin et al. (2017); Andrychowicz et al. (2020) randomizes the parameters in source environments to approximate the potential noise in task environments. Thirdly, adversarial attack approaches Huang et al. (2017); Mandlekar et al. (2017) utilize attacks with policy gradient to adversarially enhance system robustness. However, both DR and adversarial attack explicitly involve or model the observation noise during the training process. In real-world tasks, there exists non-stationary noise which is hard to model precisely, or even intractable to model Karra & Karim (2009). Therefore, prior works only work with stationary noise scenarios. Moreover, these extra noise modeling procedures often induce observable learning efficiency decrease.

## 2.3 Masked Multivariate Time Series Modeling

Masked Language Modeling (MLM) has achieved significant success in NLP domain Radford et al. (2018; 2019). MLM masks a portion of word tokens from the input sentence and trains the model to predict the masked tokens, which has been demonstrated to be generally effective in learning language representations for various downstream tasks. For Computer Vision (CV) tasks, Masked Image Modeling (MIM) learns representations for images by pre-training neural networks to reconstruct masked pixels from visible ones Vincent et al. (2008); He et al. (2022). In model-based reinforcement learning domain, Yu et al. (2022); Seo et al. (2023b;a) introduce masked-based latent reconstruction methods to enhance the learning efficiency of DRL algorithms that use images as input. This paper explores masked multivariate time series modeling for DRL to improve the robustness against observation noise.

## 3 De-noising constRained Policy Optimization Problem (DRPOP)

Locomotion control problems can be generally formulated as a MDP defined by the tuple $\langle \boldsymbol{S}, \boldsymbol{A}, \boldsymbol{P}, \rho_0, r, \gamma, T \rangle$. Where $\boldsymbol{S}$ is ground truth state. $\boldsymbol{A}$ is set of agent actions, which is taken by policy $\pi(a \mid s)$. $\boldsymbol{P} : \boldsymbol{S} \times \boldsymbol{A} \times \boldsymbol{S} \mapsto [0, 1]$ is system transition probability distribution. $\rho_0 : \boldsymbol{S} \mapsto [0, 1]$ is initial state distribution. $r : \boldsymbol{S} \times \boldsymbol{A} \times \boldsymbol{S} \mapsto \mathbb{R}$ is reward function. $\gamma$ is the discount factor where $\gamma \in [0, 1]$, and $T$ is episode horizon.

Due to the objective presence of stationary and non-stationary observation noise in environment, we formalize the process of noise affecting environment observations as $\boldsymbol{\zeta}(s) : \boldsymbol{S} \to \boldsymbol{S_{/n}}$. Under this particular disturbed condition, the agent takes an action from policy $\pi(a \mid \boldsymbol{\zeta}(s))$. Because observation procedure does not affect with objective world, the environment still transits from the ground truth state $s$ rather than $\boldsymbol{\zeta}(s)$ to the next state. Since there exist remarkable gaps between $\boldsymbol{\zeta}(s)$ and $s$, the actions be taken under $\pi(a \mid \boldsymbol{\zeta}(s))$ will gradually deviate from the optimal trajectory over time. Therefore, it's critical to optimally estimate the true state from noisy observations $\boldsymbol{\eta}(s_{/n}) : \boldsymbol{S_{/n}} \to \hat{\boldsymbol{S}}$. This paper proposes a de-noising process for this state estimation task, which aims to minimize the difference between state estimations under noise and ground truth. We expand the overall de-noising MDP process trajectory as below:

$$
\begin{aligned}
\pi_\theta(\hat{\tau}) &= \pi_\theta \left( \boldsymbol{\eta}(\boldsymbol{\zeta}(s_1)), a_1, \ldots, \boldsymbol{\eta}(\boldsymbol{\zeta}(s_T)), a_T \right) \\
&= p(s_1) \prod_{t=1}^{T} \pi_\theta \left( a_t \mid \boldsymbol{\eta}(\boldsymbol{\zeta}(s_t)) \right) p(s_{t+1} \mid s_t, a_t).
\end{aligned}
\tag{1}
$$

Subsequently, to convert the above de-noising procedure into an optimal representation learning form, we present this state estimation deviation minimization process as **D**e-noising const**R**ained **P**olicy **O**ptimization **P**roblem (**DRPOP**). Concretely, the de-noising process has to ensure the deviation between their probability distributions bounded by an acceptable small $\epsilon$, and we can achieve this by regularizing:

$$D_{KL}\left[p\left(\boldsymbol{\eta}(\boldsymbol{\zeta}(s))\right)\|p(s)\right] < \epsilon. \tag{2}$$

Thereby, this particular auxiliary regularizer can be modeled as a self-supervised learning style as:

$$\mathcal{J}(\eta) = \min \mathbb{E}_{(\hat{s})\sim\boldsymbol{\eta}(\boldsymbol{\zeta}(s))}\|\hat{s} - s\|_p. \tag{3}$$

Overall, the objective of DRPOP is to maximize the expected cumulative discounted reward of the original MDP based on this self-supervised learning style optimal state representation $\boldsymbol{\eta}(\boldsymbol{\zeta}(s))$:

$$\mathcal{J}(\pi_\theta) = \max \mathbb{E}_{(\boldsymbol{s},\boldsymbol{a})\sim\rho_{\pi_\theta}(\boldsymbol{\eta}(\boldsymbol{\zeta}(s)),\boldsymbol{a})}\left[\sum_{k=0}^{T}\gamma^k r\left(s_{t+k}, a_{t+k}\right)\right]. \tag{4}$$

According to the policy gradient theorem proposed by Sutton et al. (1999), the gradient of $\mathcal{J}\left(\pi_\theta\right)$ w.r.t $\theta$ can be derived as below (See Appendix A for more details):

$$\nabla_\theta\mathcal{J}(\pi_\theta) \approx \frac{1}{N}\sum_{n=1}^{N}\sum_{t=1}^{T_n}R\left(\boldsymbol{s}^n, \boldsymbol{a}^n\right)\nabla\log\pi_\theta\left(a_t^n \mid \boldsymbol{\eta}(\boldsymbol{\zeta}(s_t^n))\right). \tag{5}$$

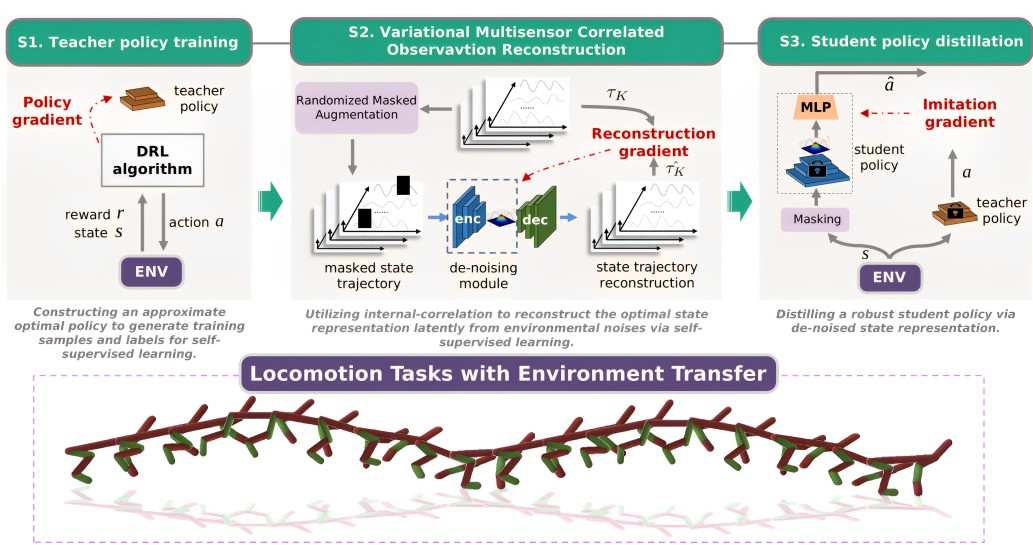

Figure 2: Architecture of SIMA.

## 4 METHODOLOGY

### 4.1 OVERVIEW

Specifically, **S**elf-supervised random**I**zed **M**asked **A**ugmentation (**SIMA**) can be categorized as a three-stage pipeline, including (S1) teacher policy training, (S2) variational multi-sensor correlated observations reconstruction, and (S3) student policy distillation, (as shown in Figure 2). Firstly, SIMA constructs a teacher policy to generate training samples and labels for the upcoming self-supervised learning procedure. Subsequently, SIMA utilizes the multi-sensor signals' internal-correlation to reconstruct the optimal state estimation representation latently. Accordingly, a self-supervised learning process is adopted to recover the randomly masked states. Consequently, a student policy, which observes the decoded latent state representation, learns a robust policy that can apply to both stationary and non-stationary environmental noise under the guidance of teacher policy via policy distillation.

## 4.2 TEACHER POLICY TRAINING

In the first stage, teacher policy observes the ground truth state from the environment directly. It is treated as an ordinary MDP, and can be solved by adopting off-the-shelf RL methods (e.g., Schulman et al. (2017); Haarnoja et al. (2018b); Fujimoto et al. (2018)). An agent selects an action $a$ from teacher policy $\pi_\vartheta(s)$ and receives a reward $r$ from the environment. The objective of teacher policy training is to find an optimal policy $\pi_\vartheta^*$ that maximizes the discounted sum of rewards over an infinite time horizon, which is $\mathcal{J}(\pi_\vartheta) = \max \mathbb{E}_{(s,a)\sim\rho_{\pi_\vartheta}}\left[\sum_{k=0}^T \gamma^k r\left(s_{t+k}, a_{t+k}\right)\right]$. The gradient of $\mathcal{J}\left(\pi_\vartheta\right)$ w.r.t $\vartheta$ is $\nabla_\vartheta \mathcal{J}\left(\pi_\vartheta\right) = \mathbb{E}_{(s,a)\sim\rho_{\pi_\vartheta}}\left[\nabla_\vartheta \log \pi_\vartheta(s,a) Q^{\pi_\vartheta}(s,a)\right]$, where $Q^{\pi_\vartheta}(s,a) = \mathbb{E}_{\pi_\vartheta}\left[\sum_{k=0}^T \gamma^k r\left(s_t, a_t\right) \mid s_0 = s, a_0 = a\right]$ is the state-action value function. The training samples $x$ are stored and reused by the upcoming self-supervised learning process.

## 4.3 VARIATIONAL MULTI-SENSOR CORRELATED OBSERVATION RECONSTRUCTION

In the second stage, we focus on learning the internal-correlation of multivariate time series via a self-supervised learning process, i.e., *variational multi-sensor correlated observation reconstruction*. The training sample $x$ is generated by the teacher policy training process. *Variational multi-sensor correlated observation reconstruction* consists of *randomized masked augmentation* and *masked state observation reconstruction*. Concretely, *randomized masked augmentation* continuously generates augmented state observations using masks, thereby promoting the training process of *masked state observation reconstruction*.

**Randomized Masked Augmentation**. Vincent et al. (2008) reveals that adding masks to the argument makes the generative model have good fill-in-the-blanks performance, which is beneficial to better capture the distribution of the argument. Given a state trajectory of $K$ timesteps $\tau_K = \{x_t, x_{t+1}, \cdots, x_{t+K-1}\}$, which starts at time $t$ and ends at time $t + K - 1$. Each state $x$ represents a $N$-dimensional variable. All the states in the trajectory are stacked to be a sequence with the shape of $K \times N$. To enhance the learning of *masked state observation reconstruction*, we introduce a novel mask mechanism named *randomized masked augmentation*

, which is empirically designed as a type of temporal sequence mask strategy. To begin with, a mask token with the width of $k$ timesteps is constructed which is denoted by $\mathcal{M}_{tok}$. Then, we create a vector $\mathcal{M}_{Ber}$ with the shape of $1 \times N$. Each element $B$ of $\mathcal{M}_{Ber}$ follows a Bernoulli distribution $Bernoulli(1, p)$. We define an indicator function $\mathbb{I}_1(B_n)$ to represent whether there is a mask in the $n_{th}$ dimension. If $\mathbb{I}_1(B_n) = 1$, we randomly select a moment $l$ on the $n_{th}$-dimensional state trajectory as the origin of the mask token and $l$ follows the uniform distribution $Uniform(t, t + K - 1)$. Therefore, a state observation trajectory $\tilde{\tau}_K =$

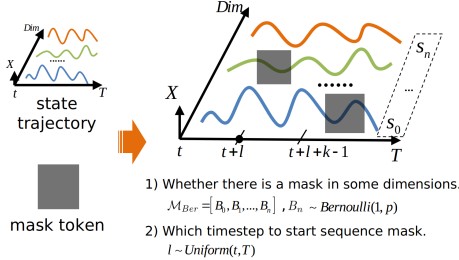

Figure 3: Illustration of *randomized masked augmentation*.

$\{x_t, \cdots, \tilde{x}_{t+l}, \tilde{x}_{t+l+1}, \cdots, \tilde{x}_{t+l+k-1}, \cdots, x_{t+K-1}\}$ has been generated, which contains temporal sequence masks with the length of $k$. The $n_{th}$ dimension of $\tilde{x}_{t+l}$ to $\tilde{x}_{t+l+k-1}$ is replaced by mask token as illustrated in Figure 3. Finally, we also added some small amplitude high-frequency noise to further enhance the de-noising ability of the DRL agent. Essentially, due to the invariance of feature parameters, *randomized masked augmentation* (denoted as $r_\psi(\tilde{x} \mid x)$) obeys a deterministic probability distribution with a theoretical bound (see Lemma 1 for proof details).

**Lemma 1** *Randomized masked augmentation subjects to a deterministic probability distribution $r_\psi(\tilde{x} \mid x)$, denoted as $\tilde{x}_i \sim r_\psi(\tilde{x} \mid x)$. Then an evidence lower bound of marginal likelihood for the argument $x$ exists:*

$$\log P_\theta(\boldsymbol{x}) = \log \int_{\tilde{\boldsymbol{x}}} P_\theta(\tilde{\boldsymbol{x}}, \boldsymbol{x}) d\tilde{\boldsymbol{x}} \qquad (6)$$

$$\inf_{\tilde{\boldsymbol{x}}_i \sim r_\psi(\tilde{\boldsymbol{x}}|\boldsymbol{x})} \log \int_{\tilde{\boldsymbol{x}}} P_\theta(\tilde{\boldsymbol{x}}, \boldsymbol{x}) d\tilde{\boldsymbol{x}} := \frac{1}{N} \sum_{i=1}^{N} [\log P_\theta(\tilde{\boldsymbol{x}}_i, \boldsymbol{x}) - \underbrace{\log r_\psi(\tilde{\boldsymbol{x}}_i \mid \boldsymbol{x})}_{C_i}] \tag{7}$$

*Proof. See Appendix B for proof details.*

**Masked State Observation Reconstruction**. *Masked state observation reconstruction* aims to reconstruct the optimal state representation from the masked state time series. The objective function is the maximum likelihood estimation of state argument $\boldsymbol{x}$, in the form of $\mathcal{J}(\theta) = \max \mathbb{E}_{\boldsymbol{x} \sim \boldsymbol{D}} [\log p_\theta(\boldsymbol{x})]$, where data set $\boldsymbol{D}$ is generated by the teacher policy. The generative model adopted in this work is Variational Auto-Encoder (VAE) Kingma & Welling (2013). The prior distribution of VAE's latent variables is set as multivariate normal distribution $p(\boldsymbol{z}) := \mathcal{N}(\boldsymbol{\mu}, \boldsymbol{\sigma}^2)$, where $\boldsymbol{\mu}$ is the mean value and $\boldsymbol{\sigma}$ is the standard deviation. A lstm-based encoder $q_\phi(\boldsymbol{z} \mid \tilde{\boldsymbol{x}})$ is adopted to encode masked state observation into the latent state representation. We treat this encoder as a de-noising module. Afterward, we build the decoder $p_\theta(\boldsymbol{x} \mid \boldsymbol{z})$ to learn to reconstruct state observation from latent state representation. The training process is self-supervised, and the state truth trajectory $\tau_K$ itself serves as the training label. According to Theorem 1, a variational evidence lower bound of the state reconstruction $\boldsymbol{x}$ exists. By continuously optimizing and elevating variational evidence lower bound, *masked state observation reconstruction* achieves the objective of learning latent state representations from masked observations seamlessly.

**Theorem 1** *Based on Lemma 1, we define the encoder and decoder of VAE as $q_\phi(\boldsymbol{z} \mid \tilde{\boldsymbol{x}})$ and $p_\theta(\boldsymbol{x} \mid \boldsymbol{z})$, respectively. Assuming that the prior distribution of latent variables obeys multivariate normal $p(\boldsymbol{z}) := \mathcal{N}(\boldsymbol{\mu}, \boldsymbol{\sigma}^2)$, where $\boldsymbol{\mu}$ is the mean value and $\boldsymbol{\sigma}$ is the standard deviation. Then a variational evidence lower bound of marginal likelihood for the argument $\boldsymbol{x}$ exists:*

$$\inf_{\boldsymbol{x} \sim P_\theta(\boldsymbol{x})} \inf_{\tilde{\boldsymbol{x}}_i \sim r_\psi(\tilde{\boldsymbol{x}}|\boldsymbol{x})} \log \int_{\tilde{\boldsymbol{x}}} P_\theta(\tilde{\boldsymbol{x}}, \boldsymbol{x}) d\tilde{\boldsymbol{x}} :=$$
$$E_{\tilde{\boldsymbol{x}}_i \sim r_\psi(\tilde{\boldsymbol{x}}|\boldsymbol{x})} \left[ E_{q_\phi(\boldsymbol{z}|\tilde{\boldsymbol{x}}_i)} [\log P_\theta(\tilde{\boldsymbol{x}}_i, \boldsymbol{x} \mid \boldsymbol{z})] - D_{KL} [q_\phi(\boldsymbol{z} \mid \tilde{\boldsymbol{x}}_i) \| P_\theta(\boldsymbol{z})] \right] - \frac{1}{N} \sum_{i=1}^{N} C_i, \tag{8}$$

*where $\psi$ is constant, $\psi \perp\!\!\!\perp \theta$, $\psi \perp\!\!\!\perp \phi$. When sample quantity $N \to +\infty$, $\frac{1}{N} \sum_{i=1}^{N} C_i$ is constant.*
*Proof. See Appendix C for all proof details.*

### 4.4 STUDENT POLICY DISTILLATION.

The aforementioned teacher policy $\pi_\vartheta$ is adopted to guide the learning process of student policy $\pi_s$, which is constructed based on the de-noising module $e_\phi$ and MLP $\pi_\omega$. Notably, the parameters of both teacher policy and de-noising module are inherited from the preceding stages and remain unchanged. We employ dataset aggregation strategy Ross et al. (2011) to distill the student policy. The teacher policy $\pi_\vartheta$ is employed to collect a dataset $\mathcal{D}_1$ of trajectories at the first iteration, which serves as the training data for the student policy. Each visited state $s$ will be randomized masked according to *randomized masked augmentation* and transmitted to the student policy as observation $s_{/m}$, i.e., $\chi(s) : \boldsymbol{S} \to \boldsymbol{S}_{/m}$. The latent state representation is encoded by the de-noising module $z = e_\phi(s_{/m})$ and the MLP outputs action $a = \pi_\omega(z)$. The action vectors $\hat{a} = \pi_\vartheta(s)$ from the teacher policy are used as supervisory signals associated with the corresponding state. The objective function of the student policy is defined by $\mathcal{J}(\omega) = \min_\omega \mathbb{E}_{(s,a) \sim \rho_{(\pi_\vartheta, \pi_s)}} (\pi_\omega(e_\phi(\chi(s))) - \pi_\vartheta(s))^2 = \min_\omega \mathbb{E}_{(s,a) \sim \rho_{\pi_i}} (\pi_s(s_{/m}) - \pi_\vartheta(s))^2$. Afterward, we use $\pi_i = \beta_i \pi_\vartheta + (1 - \beta_i) \pi_s$ to collect more trajectories at iteration $i$, and add them to the dataset $\mathcal{D} \leftarrow \mathcal{D} \cup \mathcal{D}_i$, where $\beta_i$ is the weight coefficient of sampling policy and $\beta_i \in [0, 1]$. The above process is repeatedly executed until the student policy is trained to convergence.

## 5 EXPERIMENTS

In this section, we first describe the experimental settings, then provide comprehensive experimental results to answer the following research questions:

**RQ1:** How does the proposed SIMA algorithm perform by comparing with state-of-the-art methods under environmental observation noise?

**RQ2:** Can SIMA maintain outstanding state truth distribution reconstruction performance under disturbance?

**RQ3:** Can SIMA still work when encountering both stationary and non-stationary environmental observation noise with unseen distributions?

**RQ4:** Does the performance of SIMA suffer from a notable drop when removing or changing any crucial component from the approach?

**RQ5:** Does SIMA have the potential to enhance robustness of different DRL baselines?

**RQ6:** Does SIMA outperform state-of-the-art masked model-based RL method, e.g., Mask World Models (MWM)?

**RQ7:** Does SIMA outperform robotics locomotion state representation learning methods?

Six locomotion control tasks from Pybullet Coumans & Bai (2016) are adopted. The experimental results and in-depth analysis of RQ1, RQ2, and RQ3 are shown subsequently. Beyond this, to fully demonstrate the effectiveness of SIMA from multiple perspectives, we further provide four additional experiments in appendixes. Specifically, Appendix G is for RQ4; Appendix H is for RQ5; Appendix I is for RQ6; Appendix J is for RQ7.

## 5.1 LEARNING PERFORMANCE UNDER ENVIRONMENTAL OBSERVATION NOISE (RQ1)

In this section, we compare SIMA with several SOTA methods including RL-vanilla, RL-lstm, RL-filter, and RL-Domain Randomization (RL-DR) to answer RQ1. Six typical locomotion control environments from Pybullet are adopted. Additionally, we add both stationary and non-stationary observation noise in evaluation environments (see more details in section 5.3). PPO Schulman et al. (2017) is selected as implementation backbone for all methods.

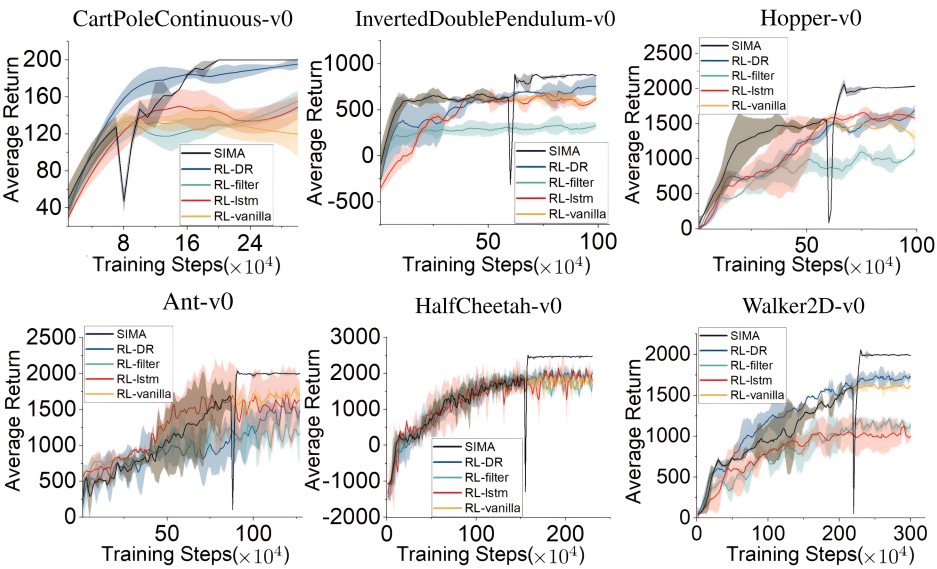

Figure 4: Learning curves.

Figure 4 shows their learning curves in terms of average return. SIMA converges in the leading edge, and shows outstanding performance in learning ability. It's worth noting that SIMA consists of three training stages. In the first two stages, SIMA utilizes clean observations to train a teacher policy and a de-noising module. Due to the fact that the teacher policy of SIMA agent in the first two stages was trained without environmental observation noise, it performs similar to other algorithms in the first half of the training curves as illustrated in Figure 4. In the last stage, we initialize

a student policy network to complete robust locomotion control tasks in noisy environments, which utilizes optimal state representations learned in the first two stages. Since the student policy lacks of guidance from the teacher policy in the first two stages, a notable performance drop pops-up in the middle section of the learning curves as shown in Figure 4. Consequently, after a short warm-up period intervened by the teacher policy, the performance of SIMA outperforms other algorithms. To give a clearer description of the three stages of SIMA, we provide a schematic diagram in Figure 8 in Appendix E. On the other hand, it is observed that SIMA performs well whether in terms of low or high observation dimensions, while other baselines perform relatively weak dueing to uncertainty induced by noise increasing in high-dimensional environments. In contrast, SIMA learns the correlation of multivariate time series via self-supervised learning without modeling environmental noise. Therefore, negative impact of learning efficiency with high-dimensional observations is reduced to a certain extent. In summary, SIMA outperforms SOTA methods by **15.7**% increase in learning performance. Besides the learning curves, we also evaluate these agents with four types of noisy scenes by 30 trials (As shown in Appendix F), and SIMA is still proven to have notable advantages.

## 5.2 MAINTAINING OUTSTANDING STATE TRUTH DISTRIBUTION RECONSTRUCTION PERFORMANCE UNDER DISTURBANCE (RQ2)

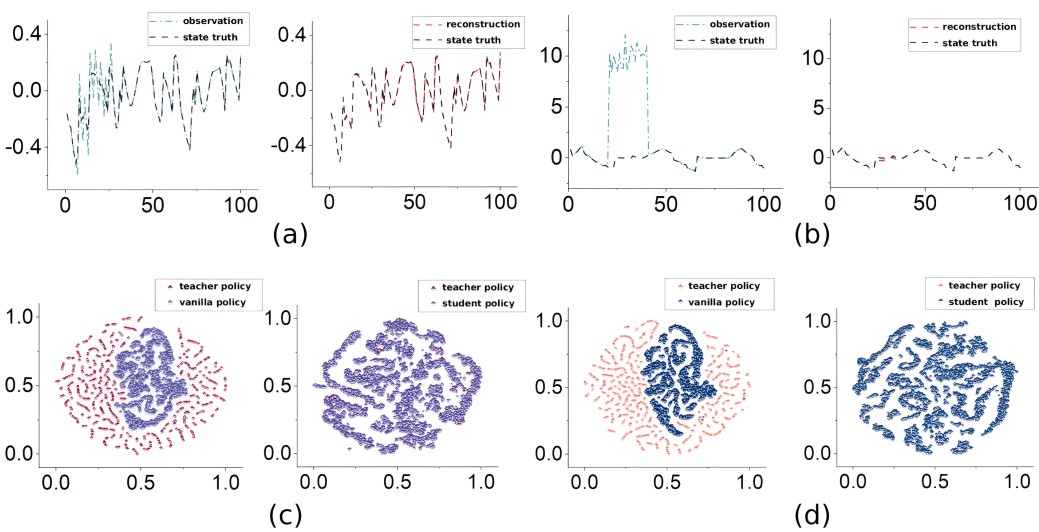

Figure 5: Analysis of (a), (b) state observation reconstruction, and (c), (d) policy distribution.

This section analyzes state truth distribution reconstruction performance under disturbance to answer RQ2. We design two experiments under stationary and non-stationary environmental noise as below: Firstly, we conduct an experiment to test de-noising performance of SIMA. This particular experiment adopts one-dimensional state observation (i.e., leg joint angular velocity of HalfCheetah) as system input. The results are shown in Figure 5, in which Figure 5 (a) shows state recovery result with stationary noise, and Figure 5 (b) shows the one with non-stationary noise. It is worth noting that there exists a clear-cut non-stationary noise phase in Figure 5 (b). In such case, it can be seen that SIMA properly reconstructs the correct signal from this unexpectable disturbance. In view of this, we demonstrate that SIMA significantly enhances the overall system robustness against environmental noise. Secondly, to further demonstrate that SIMA can also accurately reconstruct the correct state observations from noises. We trained a teacher policy with state truth as a reference (illustrated as red dots scattered in Figure 5), a student policy to reconstruct the optimal policy embeddings under these unexpectable disturbance, and a vanilla DRL policy to serve as the control group. All these parts are trained with $2 \times 10^6$ samples. We utilize t-SNE to visualize the corresponding distributions of the three policies in Figure 5, where 5 (c) represents experiment with stationary noise, and Figure 5 (d) represents the one with non-stationary noise. It can be clearly seen that the distribution of the student policy almost completely reconstructed the policy space of the teacher policy in both disturbance conditions. Besides, the distribution of the vanilla DRL policy shows a significant deviation from the teacher policy. These experimental results further support that SIMA

can accurately estimate and reconstruct state representation from disturbance, and enhance system robustness significantly.

## 5.3 Generalization Analysis under various Stationary and Non-stationary Observation noise conditions(RQ3)

Generalization is an important metric for deep learning algorithms. In this experiment, we evaluate if SIMA can be generalized to the environments with various unseen environmental noise. Specifically, we adopt both stationary noise (high-frequency noise) and non-stationary noise (intermittent disturbance), each with 9 groups of different noise characteristics. The stationary noise groups follow gaussian process with frequency mean as $f = [16, 32, 64]$ (khz), and amplitude variance as $A = [0.4, 0.7, 1.0]$. The corresponding results are illustrated in Figure 6 (a). Alternatively, the non-stationary noise groups are set to be unpredictable in occurance and duration periods. In such case, we name it as intermittent disturbance which follows uniform distribution with duration period characteristics as $T = [10, 15, 20]$ (steps), and amplitude characteristics as $A = [5, 10, 15]$. All these noise durations occur in range of $0$ to $T$ randomly as illustrated in Figure 6 (b). All experimental results are collected from HalfCheetah environment. It can be seen that SIMA outperforms RL-vanilla by $95\%$ in task success rate (we define task success once an agent never falls down during running). Furthermore, SIMA also performs significantly smaller variance in average return than RL-vanilla, which indicates SIMA maintains more stable under different disturbance characteristics. Consequently, all these results prove that SIMA can be generalized to the environments with various unseen environmental noise.

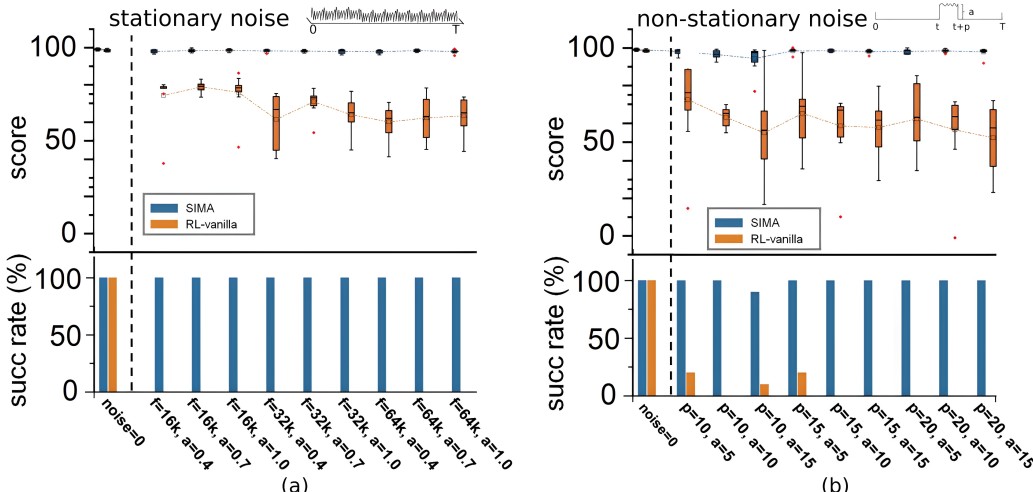

Figure 6: Analysis of generalization with (a) stationary noise and (b) non-stationary noise.

## 6 Conclusion and future work

Learning to adapt to environmental observation noise is critical to DRL-based locomotion control tasks. Since there exists non-stationary noise hardly to model precisely, or even intractable to model Karra & Karim (2009). Prior works often fail to effectively cope with the above situations. In this paper, we describe an approach for learning robust locomotion skills without modeling environmental observation noise explicitly. Inspired by *multi-sensory integration* mechanism, we first formulate the MDP with an environmental de-noising process as a DRPOP problem. On this basis, we propose a novel SIMA algorithm to accurately construct the latent state representation of ground truth from noisy observations. In essence, SIMA is a paradigm shift that significantly improves robustness of DRL agents against observation noise without explicitly modeling procedure. Experiments reveal that SIMA learns the correlation of multivariate time series, and provides a feasible path to solve the problem of DRL-based locomotion control with environmental observation noise. In future investigations, SIMA can be deployed in real-world robot applications, e.g., autonomous self-driving cars, unmanned aircraft vehicles, and any other real-world tasks.

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

# Appendix

## A    Derivation of DRPOP's objective fuction's gradient

The goal of DRPOP is to maximize the expected cumulative discounted reward of the locomotion control process based on $\boldsymbol{\eta}(\boldsymbol{\zeta}(s))$:

$$\mathcal{J}(\pi_\theta) = \max \mathbb{E}_{(\boldsymbol{s},\boldsymbol{a}) \sim \rho_{\pi_\theta}(\boldsymbol{\eta}(\boldsymbol{\zeta}(s)),\boldsymbol{a})} \left[ \sum_{k=0}^{T} \gamma^k r\left(s_{t+k}, a_{t+k}\right) \right]. \tag{9}$$

We define $\hat{\boldsymbol{s}} := \boldsymbol{\eta}(\boldsymbol{\zeta}(s))$. We denote sampling trajectory as $\tau := (\boldsymbol{s}, \boldsymbol{a})$ and $\hat{\tau} := (\hat{\boldsymbol{s}}, \boldsymbol{a})$. According to the policy gradient theorem Sutton et al. (1999), the gradient of $\mathcal{J}(\pi_\theta)$ w.r.t $\theta$ can be derived as below:

$$
\begin{aligned}
\nabla_\theta \mathcal{J}(\pi_\theta) &= \int \nabla_\theta \pi_\theta(\hat{\boldsymbol{s}}|\boldsymbol{a}) R(\boldsymbol{s},\boldsymbol{a}) d(\boldsymbol{s},\boldsymbol{a}) \\
&= \sum_{(\boldsymbol{s},\boldsymbol{a})} R(\boldsymbol{s},\boldsymbol{a}) \nabla \pi_\theta(\hat{\boldsymbol{s}}|\boldsymbol{a}) \\
&= \sum_{(\boldsymbol{s},\boldsymbol{a})} R(\boldsymbol{s},\boldsymbol{a}) \pi_\theta(\hat{\boldsymbol{s}}|\boldsymbol{a}) \frac{\nabla \pi_\theta(\hat{\boldsymbol{s}}|\boldsymbol{a})}{\pi_\theta(\hat{\boldsymbol{s}}|\boldsymbol{a})} \\
&= \sum_{(\boldsymbol{s},\boldsymbol{a})} R(\boldsymbol{s},\boldsymbol{a}) \pi_\theta(\hat{\boldsymbol{s}}|\boldsymbol{a}) \nabla \log \pi_\theta(\hat{\boldsymbol{s}}|\boldsymbol{a}) \\
&= \mathbb{E}_{(\boldsymbol{s},\boldsymbol{a}) \sim \pi_\theta(\hat{\boldsymbol{s}}|\boldsymbol{a})} \left[ R(\boldsymbol{s},\boldsymbol{a}) \nabla \log \pi_\theta(\hat{\boldsymbol{s}}|\boldsymbol{a}) \right] \\
&\approx \frac{1}{N} \sum_{n=1}^{N} R\left(\boldsymbol{s}^n, \boldsymbol{a}^n\right) \nabla \log \pi_\theta\left(\boldsymbol{a}^n \mid \boldsymbol{\eta}(\boldsymbol{\zeta}(\boldsymbol{s}^n))\right) \\
&= \frac{1}{N} \sum_{n=1}^{N} \sum_{t=1}^{T_n} R\left(\boldsymbol{s}^n, \boldsymbol{a}^n\right) \nabla \log \pi_\theta\left(a_t^n \mid \boldsymbol{\eta}(\boldsymbol{\zeta}(s_t^n))\right)
\end{aligned} \tag{10}
$$

## B    Proof of Lemma 1

Randomized masked augmentation subjects to a deterministic probability distribution $r_\psi(\tilde{\boldsymbol{x}} \mid \boldsymbol{x})$, denoted as $\tilde{\boldsymbol{x}}_i \sim r_\psi(\tilde{\boldsymbol{x}} \mid \boldsymbol{x})$. Then an evidence lower bound of marginal likelihood is proven as listed below:

$$
\begin{aligned}
\log P_\theta(\boldsymbol{x}) &= \log \int_{\tilde{\boldsymbol{x}}} P_\theta(\tilde{\boldsymbol{x}}, \boldsymbol{x}) d\tilde{\boldsymbol{x}} \\
&= \log \int_{\tilde{\boldsymbol{x}}} \frac{P_\theta(\tilde{\boldsymbol{x}}, \boldsymbol{x})}{r_\psi(\tilde{\boldsymbol{x}} \mid \boldsymbol{x})} r_\psi(\tilde{\boldsymbol{x}} \mid \boldsymbol{x}) d\tilde{\boldsymbol{x}} \\
&= \log E_{r_\psi(\tilde{\boldsymbol{x}}|\boldsymbol{x})} \left[ \frac{P_\theta(\tilde{\boldsymbol{x}}, \boldsymbol{x})}{r_\psi(\tilde{\boldsymbol{x}} \mid \boldsymbol{x})} \right]
\end{aligned} \tag{11}
$$

According to Jensen's inequality,

$$\log P_\theta(\boldsymbol{x}) \geq E_{r_\psi(\tilde{\boldsymbol{x}}|\boldsymbol{x})} \log \left[ \frac{P_\theta(\tilde{\boldsymbol{x}}, \boldsymbol{x})}{r_\psi(\tilde{\boldsymbol{x}} \mid \boldsymbol{x})} \right]$$

$$= E_{r_\psi(\tilde{\boldsymbol{x}}|\boldsymbol{x})} \left[ \log P_\theta(\tilde{\boldsymbol{x}}, \boldsymbol{x}) - \log r_\psi(\tilde{\boldsymbol{x}} \mid \boldsymbol{x}) \right]$$

$$= \frac{1}{N} \sum_{i=1}^{N} [\log P_\theta(\tilde{\boldsymbol{x}}_i, \boldsymbol{x}) - \underbrace{\log r_\psi(\tilde{\boldsymbol{x}}_i \mid \boldsymbol{x})}_{C_i}] \qquad (12)$$

$$= \frac{1}{N} \sum_{i=1}^{N} [\log P_\theta(\tilde{\boldsymbol{x}}_i, \boldsymbol{x}) - C_i]$$

$$= ELBO_{1st}$$

Because of $\psi$ is constant and $\psi \perp\!\!\!\perp \theta$, $\frac{1}{N} \sum_{i=1}^{N} C_i$ is constant, when the sample quantity $N \to +\infty$.

$$ELBO_{1st} = \frac{1}{N} \sum_{i=1}^{N} \underbrace{[\log P_\theta(\tilde{\boldsymbol{x}}_i, \boldsymbol{x})]}_{O_i} - \frac{1}{N} \sum_{i=1}^{N} C_i \qquad (13)$$

## C  PROOF OF THEOREM 1

Based on Lemma 1, we involve the latent variable $\boldsymbol{z}$.

$$O_i = \log P_\theta(\tilde{\boldsymbol{x}}_i, \boldsymbol{x})$$

$$= \log \int_{\boldsymbol{z}} P_\theta(\tilde{\boldsymbol{x}}_i, \boldsymbol{x}, \boldsymbol{z}) \, d\boldsymbol{z}$$

$$= \log \int_{\boldsymbol{z}} \frac{P_\theta(\tilde{\boldsymbol{x}}_i, \boldsymbol{x}, \boldsymbol{z})}{q_\phi(\boldsymbol{z} \mid \tilde{\boldsymbol{x}}_i)} q_\phi(\boldsymbol{z} \mid \tilde{\boldsymbol{x}}_i) \, d\boldsymbol{z} \qquad (14)$$

$$= \log E_{q_\phi(\boldsymbol{z}|\tilde{\boldsymbol{x}}_i)} \left[ \frac{P_\theta(\tilde{\boldsymbol{x}}_i, \boldsymbol{x}, \boldsymbol{z})}{q_\phi(\boldsymbol{z} \mid \tilde{\boldsymbol{x}}_i)} \right]$$

According to Jensen's inequality,

$$O_i \geq E_{q_\phi(\boldsymbol{z}|\tilde{\boldsymbol{x}}_i)} \log \left[ \frac{P_\theta(\tilde{\boldsymbol{x}}_i, \boldsymbol{x}, \boldsymbol{z})}{q_\phi(\boldsymbol{z} \mid \tilde{\boldsymbol{x}}_i)} \right]$$

$$= E_{q_\phi(\boldsymbol{z}|\tilde{\boldsymbol{x}}_i)} \left[ \log P_\theta(\tilde{\boldsymbol{x}}_i, \boldsymbol{x} \mid \boldsymbol{z}) + \log \frac{P_\theta(\boldsymbol{z})}{q_\phi(\boldsymbol{z} \mid \tilde{\boldsymbol{x}}_i)} \right] \qquad (15)$$

$$= E_{q_\phi(\boldsymbol{z}|\tilde{\boldsymbol{x}}_i)} \left[ \log P_\theta(\tilde{\boldsymbol{x}}_i, \boldsymbol{x} \mid \boldsymbol{z}) \right] + \int_{\boldsymbol{z}} q_\phi(\boldsymbol{z} \mid \tilde{\boldsymbol{x}}_i) \log \frac{P_\theta(\boldsymbol{z})}{q_\phi(\boldsymbol{z}, \tilde{\boldsymbol{x}}_i)} d\boldsymbol{z}$$

$$= E_{q_\phi(\boldsymbol{z}|\tilde{\boldsymbol{x}}_i)} \left[ \log P_\theta(\tilde{\boldsymbol{x}}_i, \boldsymbol{x} \mid \boldsymbol{z}) \right] - D_{kL} \left[ q_\phi(\boldsymbol{z} \mid \tilde{\boldsymbol{x}}_i) \| P_\theta(\boldsymbol{z}) \right]$$

So we can get a variational evidence lower bound of marginal likelihood $ELBO_{2nd}$ as below:

$$ELBO_{2nd} = E_{\tilde{\boldsymbol{x}}_i \sim r_\psi(\tilde{\boldsymbol{x}}|\boldsymbol{x})} \left[ E_{q_\phi(\boldsymbol{z}|\tilde{\boldsymbol{x}}_i)} \left[ \log P_\theta(\tilde{\boldsymbol{x}}_i, \boldsymbol{x} \mid \boldsymbol{z}) \right] - D_{KL} \left[ q_\phi(\boldsymbol{z} \mid \tilde{\boldsymbol{x}}_i) \| P_\theta(\boldsymbol{z}) \right] \right] - \frac{1}{N} \sum_{i=1}^{N} C_i \qquad (16)$$

where $\psi$ is constant, $\psi \perp\!\!\!\perp \theta$, $\psi \perp\!\!\!\perp \phi$ and only $\theta$ and $\phi$ are parameters that need to be optimized. Overall, we obtain a variational evidence lower bound $ELBO_{2nd}$.

$$\log P_\theta(\boldsymbol{x}) \geq ELBO_{1st} \geq ELBO_{2nd} \qquad (17)$$

## D   EXPERIMENTAL SETTINGS

To evaluate the performance of SIMA, six locomotion control tasks (e.g., CartPoleContinuous-v0, InvertedDoublePendulum-v0, Hopper-v0, Ant-v0, Walker2D-v0, and HalfCheetah-v0) from Pybullet Coumans & Bai (2016) are adopted. The observation dimensions of the six environments gradually increase from left to right. Particularly, the observations from the evaluation environments contain stationary noise (e.g., high-frequency noise), and also non-stationary noise (e.g., intermittent disturbance), of which the distributions are unknown during training process. The training procedure of all methods is conducted on a PC with i7-11700KF CPU with Geforce RTX 3080Ti GPU.

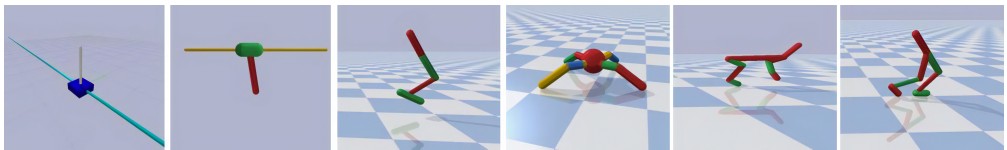

Figure 7: Pybullet experimental environments.

## E   VISUALIZATION OF THREE SPLIT TRAINING STAGES IN LEARNING CURVES

A typical learning curve of SIMA with three split training stages is shown in Figure 8. We can observe that teacher policy training (stage 1), and variational multisensor correlated observation reconstruction (stage 2) are trained subsequently in the first half of the curve. Right after these two stages finished, student policy distillation (stage 3) begins. In the last stage, we initialize a student policy network to complete robust locomotion control tasks in noisy environments, which utilizes optimal state representations learned in the first two stages. Since the student policy lacks of guidance from the teacher policy in the first two stages, a notable performance drop pops-up in the middle section of the learning curves as shown in Figure 8. After this short warm-up period, SIMA rapidly achieves the best performance.

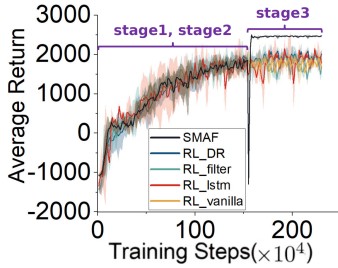

Figure 8: The schematic diagram of three stages learning performance.

## F   EVALUATION OF SIMA AND OTHER BASELINES

To evaluate the performance of post-trained SIMA agents more sufficiently, we conduct an additional assessment with five algorithms (including SIMA) in six locomotion control tasks with four different types of noise.

Firstly, we extensively train individual locomotion control policies in six clean environments (no environmental noise), and record the average returns (e.g., CartPole-200, Halfcheetah-2500). We then adopt these results as benchmarks. To map the experimental results from these different algorithms into a unified comparison range, we collect returns from 30 trials that are all normalized into values between 0.0 and 100.0 as shown in Table 1, in which *clean*, *hf-noise*, *disturbance*, and *anomaly* indicate no observation noise, high-frequency noise, intermittent disturbance, and high-frequency noise, intermittent disturbance, and outlier points all combined together, respectively. All these algorithms achieve more than 99 scores in all six *clean* environments. It can be revealed that all these

algorithms performs normally without environment noise. Results better than 90 scores are marked as bold in the table. It can also be observed that RL-filter and SIMA perform better in *hf-noise* ones. RL-DR and SIMA work well in scenarios with low-dimensional observations. Nevertheless, only SIMA maintains good performance in high-dimensional observation tasks with all disturbance settings.

Table 1: Additional performance test for SIMA and other baselines in 30 trials.

| Envs | CartPoleContinuous-v0 | | | | InvertedDoublePendulum-v0 | | | |
|---|---|---|---|---|---|---|---|---|
| Algos | clean | hf-noise | disturbance | anomaly | clean | hf-noise | disturbance | anomaly |
| RL-vanilla | **100.0 ± 0.0** | 69.6 ± 17.3 | 63.5 ± 23.7 | 62.3 ± 20.4 | **99.6 ± 0.8** | 88.3 ± 5.3 | 80.3 ± 4.7 | 85.9 ± 4.9 |
| RL-filter | **100.0 ± 0.0** | **98.9 ± 3.5** | 72.9 ± 7.2 | 76.4 ± 6.2 | **99.5 ± 1.1** | **98.8 ± 0.7** | 49.9 ± 7.7 | 52.8 ± 6.1 |
| RL-DR | **100.0 ± 0.0** | **99.3 ± 2.7** | **93.7 ± 7.3** | **97.4 ± 5.3** | **99.2 ± 1.2** | **94.6 ± 6.8** | 86.5 ± 14.8 | 87.6 ± 11.4 |
| RL-lstm | **100.0 ± 0.0** | 76.4 ± 11.7 | 67.1 ± 13.2 | 75.6 ± 12.4 | **99.6 ± 1.1** | 86.3 ± 4.3 | 82.7 ± 3.9 | 86.3 ± 5.6 |
| SMARL | **100.0 ± 0.0** | **100.0 ± 0.0** | **99.9 ± 0.1** | **99.9 ± 0.1** | **99.4 ± 1.0** | **99.5 ± 0.7** | **99.1 ± 1.3** | **99.2 ± 1.1** |
| Envs | Hopper-v0 | | | | Ant-v0 | | | |
| Algos | clean | hf-noise | disturbance | anomaly | clean | hf-noise | disturbance | anomaly |
| RL-vanilla | **99.8 ± 0.2** | 66.6 ± 3.2 | 51.7 ± 4.7 | 52.1 ± 3.1 | **99.8 ± 0.1** | 85.5 ± 7.5 | 88.9 ± 5.9 | 87.7 ± 6.6 |
| RL-filter | **99.5 ± 0.1** | **98.9 ± 0.3** | 47.9 ± 3.1 | 48.3 ± 2.4 | **99.8 ± 0.2** | **98.8 ± 0.6** | 61.7 ± 7.3 | 62.8 ± 7.7 |
| RL-DR | **99.0 ± 0.2** | 83.7 ± 4.6 | 70.2 ± 3.9 | 69.8 ± 5.7 | **99.1 ± 0.7** | **90.8 ± 9.0** | 77.1 ± 12.3 | 76.6 ± 10.0 |
| RL-lstm | **99.7 ± 0.1** | 57.3 ± 2.8 | 62.1 ± 4.0 | 62.8 ± 3.3 | **99.6 ± 0.1** | 86.9 ± 6.0 | 84.1 ± 7.1 | 85.6 ± 6.8 |
| SMARL | **99.7 ± 0.1** | **99.2 ± 0.5** | **99.3 ± 0.4** | **99.2 ± 0.6** | **99.6 ± 0.2** | **99.0 ± 1.2** | **99.6 ± 0.7** | **99.2 ± 1.7** |
| Envs | Walker2Dv0 | | | | HalfCheetah-v0 | | | |
| Algos | clean | hf-noise | disturbance | anomaly | clean | hf-noise | disturbance | anomaly |
| RL-vanilla | **99.8 ± 0.6** | 83.2 ± 5.4 | 86.1 ± 5.5 | 85.9 ± 5.3 | **99.5 ± 0.6** | 84.5 ± 5.0 | 85.7 ± 4.6 | 85.6 ± 5.2 |
| RL-filter | **99.5 ± 1.3** | **99.0 ± 5.6** | 84.8 ± 3.7 | 86.3 ± 4.1 | **99.5 ± 1.0** | **99.1 ± 2.1** | 52.3 ± 3.7 | 53.2 ± 3.9 |
| RL-DR | **99.2 ± 1.5** | 87.1 ± 9.3 | 87.0 ± 7.8 | 86.9 ± 7.1 | **98.9 ± 2.6** | 89.6 ± 5.5 | 82.8 ± 7.5 | 88.2 ± 8.3 |
| RL-lstm | **99.6 ± 1.1** | 86.4 ± 4.1 | 87.1 ± 3.3 | 87.1 ± 3.2 | **99.6 ± 0.7** | 66.3 ± 10.4 | 48.9 ± 17.2 | 49.3 ± 16.3 |
| SMARL | **99.6 ± 0.9** | **99.4 ± 0.8** | **99.4 ± 1.0** | **99.2 ± 1.4** | **99.4 ± 0.9** | **99.4 ± 0.9** | **99.1 ± 0.7** | **99.2 ± 0.6** |

# G   ABLATION STUDY (RQ4)

**Component of SIMA.** We assess the role of different parts of SIMA in this experiment. SIMA with both *randomized masked augmentation* and *masked state observation reconstruction* is trained directly as a baseline, which is compared to SIMA without *randomized masked augmentation* (denoted by SIMA-wo-mask) and SIMA without *masked state observation reconstruction* (denoted by SIMA-wo-reconstruction). Figure 9 shows that the performance of SIMA outperforms the other two variants. Since *masked state observation reconstruction* is designed to learn internal correlation between multivariate time series via self-supervised learning, *randomized masked augmentation* is designed to enrich the distribution of training data samples. It can be observed that the absence of either *randomized masked augmentation* or *masked state observation reconstruction* significantly reduce robustness of SIMA.

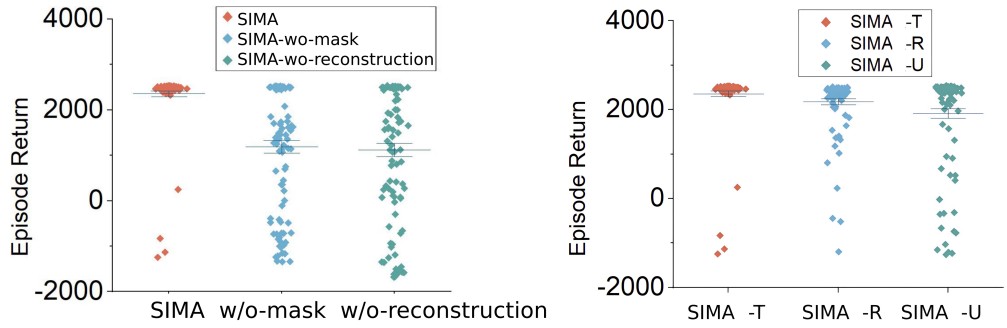

Figure 9: Ablation of SIMA components.    Figure 10: Ablation of masked strategy.

**Masked Augmentation Strategy.** We discuss the role of masked augmentation strategy in this part. Three types of *randomized masked augmentation* strategies are compared: (1) Random masking (denoted as SIMA-R), (2) Uniform masking (denoted as SIMA-U), and (3) Temporal sequence masking (denoted as SIMA-T). SIMA-T denotes that the state observation is masked with a fixed length sequence window, as illustrated in Figure 3. Figure 10 shows that SIMA-T has significant higher episodic return than the other two strategies, with smaller variance and more concentrated

Table 2: Hyperparameters of SIMA.

| Buffer size | Latent dimension | Train loss | Validation loss |
|:---:|:---:|:---:|:---:|
| 0.1M | 8 | 134.759 | 1068.834 |
| 0.1M | 16 | 8.156 | 1114.599 |
| 0.1M | 24 | 3.955 | 709.471 |
| 0.1M | 48 | 3.823 | 706.53 |
| 0.3M | 8 | 137.329 | 138.516 |
| 0.3M | 16 | 9.259 | 9.776 |
| **0.3M** | **24** | **5.037** | **5.726** |
| 0.3M | 48 | 5.355 | 6.074 |
| 0.6M | 8 | 137.657 | 142.386 |
| 0.6M | 16 | 9.223 | 9.715 |
| 0.6M | 24 | 5.182 | 5.589 |
| 0.6M | 48 | 4.979 | 6.334 |

distribution. Accordingly, it can be revealed that temporary sequence masking can help *masked state observation reconstruction* to capture the distribution of multivariate time series more precisely.

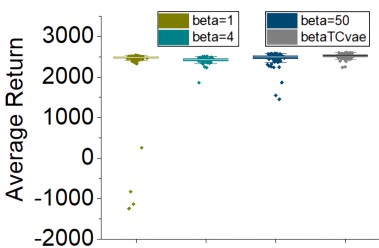

Figure 11: Different generative models.

**Generative Models.** We attempt BetaVAE Higgins et al. (2016) and Beta-TC-VAE Chen et al. (2018) as benchmark generative models to encode masked state observation into the the latent state representation. Experimental environment is Halfcheetah-v0 with observation noise. For BetaVAE model, we set three sets of parameters $\beta = 1$ (i.e., vanilla VAE), $\beta = 4$, and $\beta = 50$. For Beta-TC-VAE model, we set the parameters as $\alpha = 1$, $\beta = 1$, and $\gamma = 6$. The results can be seen in Figure 11. The four generative models have almost the same impact on SIMA performance, with both mean and variance being close. BetaVAE and Beta-TC-VAE aim to balance latent channel capacity and independence constraints with reconstruction accuracy. The locomotion control experimental scenarios in this paper belong to continuous dynamic system. Good disengagement could not result in a significant improvement in downstream continuous control tasks. Experimental results indicate that vanilla VAE can satisfy the requirements of SIMA.

**Hyperparameters of *masked state observation reconstruction*.** This experiment is a comparison of hyperparameters perturbation, i.e., buffer size, and latent representation dimension. To ensure that *masked state observation reconstruction* accurately captures the distribution of multivariate time series state observations, we empirically designed a training dataset and a validation dataset. This part is a detailed comparison of perturbation on hyperparameters, i.e., buffer size and latent dimension. In terms of buffer size, it can be seen from Table 2 that when buffer size is $3 \times 10^5$ and latent dimension is 24, training loss and validation loss reach the minimum values simultaneously. When buffer size $< 3 \times 10^5$, the validation loss becomes larger, indicating that *masked state observation reconstruction* is overfitting, and when buffer size $> 3 \times 10^5$, the training loss no longer decreases, indicating that training process involves a large number of redundant calculations. In terms of latent dimension, it can be seen that if latent dimension $< 24$, the model is underfitting. Once latent dimension $> 24$, training time will be significantly extended. Consequently, we choose buffer size $= 3 \times 10^5$ and latent dimension $= 24$ as the optimal hyperparameters of *masked state observation reconstruction*.

## H ENHANCE ROBUSTNESS OF DIFFERENT DRL BASELINES (RQ5)

In this section, SIMA is proven to enhance robustness for a wide range of RL algorithms, including on-policy DRL method (PPO Schulman et al. (2017)), and off-policy methods (SAC Haarnoja et al. (2018b), TD3 Fujimoto et al. (2018)). We adopt a Halfcheetah-v0 environment with observation noise. As illustrated in Figure 12, SIMA improves robustness of all these baselines, which indicates that SIMA is capable of enhancing DRL algorithms' robustness against environmental noise.

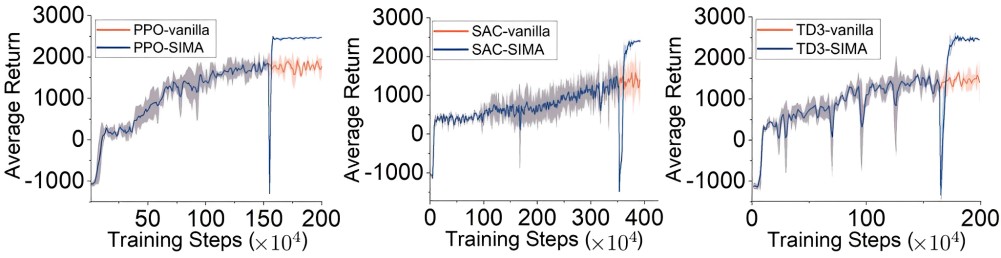

Figure 12: Robustness enhanced by SIMA for different DRL baselines.

## I COMPARATIONS WITH STATE-OF-THE-ART MASKED MODEL-BASED RL METHOD (RQ6)

In this experiment, we mainly focus on learning performance comparations between SIMA and Masked World Models (MWM) Seo et al. (2023a). The testing environment is Pybullet HalfCheetah-v0 with stationary and non-stationary observation noise. Accordingly, we change MWM's visual-input based autoencoder from convolutional layers and vision transformers (ViT) to signal-input fully connected layers and a variational autoencoder (VAE), respectively.

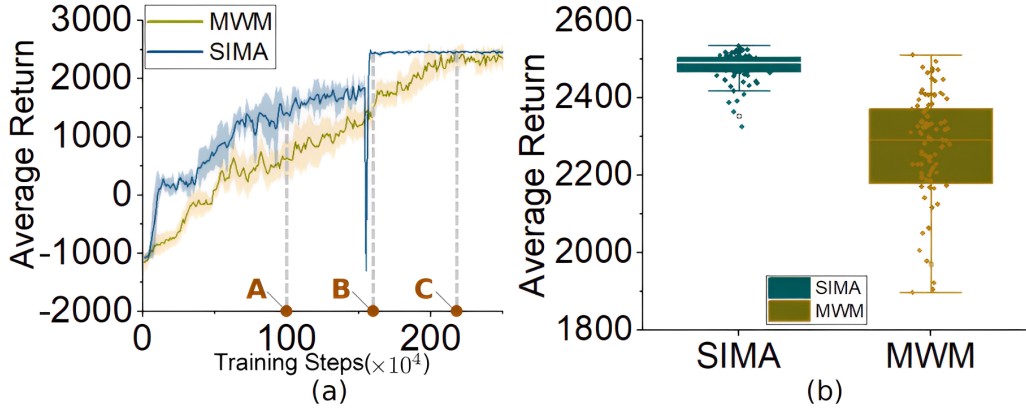

Figure 13: Evaluation of the fully trained agents.

Here, we find an interesting phenomenon in Figure 13 (a). From this figure, it can be clearly seen that SIMA shows significant improvements in learning efficiency compared to MWM in the first two training stages. We speculate the proper reason is the teacher policy training (stage 1) can directly observe the system state truth, thereby makes SIMA a quicker learner in locomotion skills. The corresponding locomotion skills bring correct state and action distributions that are critical to de-noising module learning performance in stage 2 subsequently. In contrast, MWM cannot directly observe system state truth in all training stages. In Figure 13(a), at $100 \times 10^4$ time steps (point A), it can be clearly seen that SIMA has higher learning efficiency than MWM. Afterward, we found that the SIMA algorithm shows a significant performance improvement on average returns after being connected to student policy distillation (stage 3). In Figure 13(a), at $160 \times 10^4$ time steps (point B), due to the intervention of student policy distillation (in stage3), SIMA quickly converges

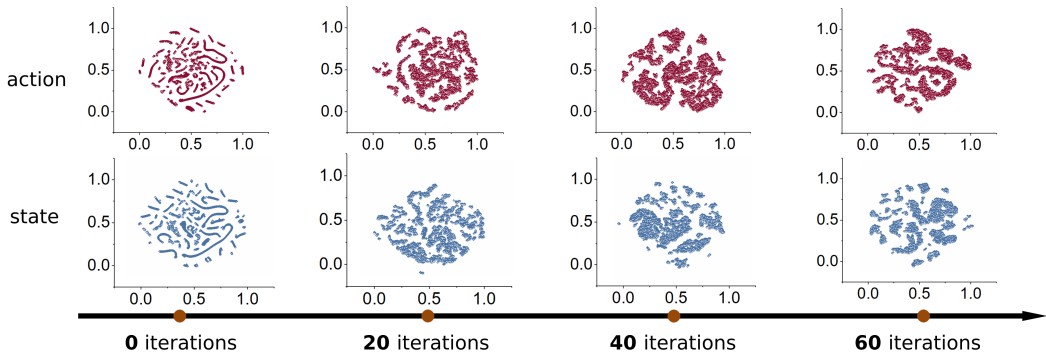

Figure 14: Changes of the sate-action pairs' distributions as the training process.

with almost no variances (2492.63±31.48). In contrast, MWM still shows significant fluctuations in scores (1387.55±227.84). SIMA outperforms MWM by 44% in training scores. MWM reaches its highest training score at $220 \times 10^4$ time steps (point C), which requires $60 \times 10^4$ samples more than SIMA. It indicates SIMA has faster convergency. We evaluated the learning performance of SIMA (trained for $160 \times 10^4$ time steps) and MWM (trained for $220 \times 10^4$ time steps). As depicted in Figure 13(b), our approach SIMA (2486.77±67.42) exhibits better learning performance than MWM (2289.43±276.33). The essential reason is that SIMA fully learned the optimal state representations under environmental noise conditions by robot running procedures during the first two stages, thus effectively suppress the environmental noise encountered.

To further demonstrate this conclusion, we conduct an additional experiment on the core differences between MWM and SIMA. In case of MWM, the sampling-training cycles of MWM last for many rounds, and in each new round, due to the drastic change of robot running policy distribution. MWM needs to re-adapt to brand new world models and de-noising modules in all training rounds. In view of this, we list the probability distributions of states and actions for multiple MWM training rounds in Figure 14. It can be clearly seen that there have been significant changes in the probability distributions of states and actions corresponding to the new trained robot's running skills in each round, resulting in unstable updates to the MWM policy. In contrast, SIMA employs a "never turn back" training mode throughout the entire training process. Once the first two stages have properly learned de-noising skills, stage 3 only needs to complete single round of student policy distillation based on this. This ensures that SIMA only needs to adapt to new de-noising probability distribution of states and actions once, and thus achieves a better learning performance. This novel learning pipeline of SIMA brings significant improvement in learning curves observed in Figure 13.

## J  COMPARATIONS WITH ROBOTICS LOCOMOTION STATE REPRESENTATION LEARNING METHODS (RQ7)

In this section, we mainly focus on comparing the performance of SIMA with other well-adopted robotics locomotion state representation learning baselines, i.e., Belief_AE Miki et al. (2022), and RMA Kumar et al. (2021). HalfCheetah-v0 environments with two types of noise are adopted.

Figure 15 and Figure 16 show experimental results under single-sensor noise conditions by assessing robot locomotion dynamics, i.e., during entire locomotion task episodes, observation noise only appears in one signal-input channel of the sensors. Figure 16 shows the average return performance among SIMA, Belief_AE, and RMA by 50 episodic trials. It can be clearly seen in Figure 16, SIMA and Belief_AE perform better than RMA. It is notable that Belief_AE performs poor in multi-sensor noise conditions, we will talk about this subsequently.

Figure 17 and Figure 18 show experimental results under multi-sensor noise conditions. Figure 18 shows that SIMA also outperforms Belief_AE by 25% in average return. More specifically, the "Belief state encoder" in Belief_AE leverages an attentional gate explicitly controls which aspects of exteroceptive data to pass through. However, the Belief_AE could handle noises only in one-demensional observation (e.g., the elevation map in Miki et al. (2022)), and thus lacks of an

important mechanism to assess the noise scale for all sensors input. Accordingly, when several different sensors suffer from noises or disturbances, Belief_AE cannot reconstruct optimal state representations correctly.

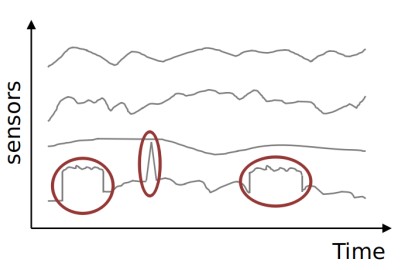

Figure 15: Diagram of single-sensor noise.

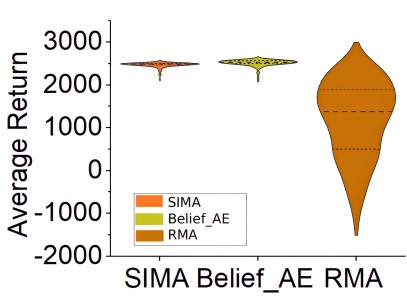

Figure 16: Reward statistics under single-sensor noise.

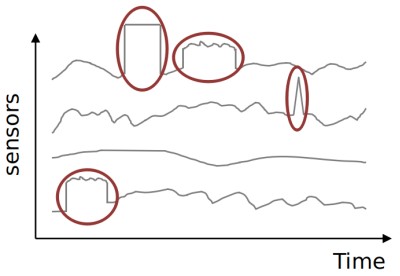

Figure 17: Diagram of multiple-sensor noise.

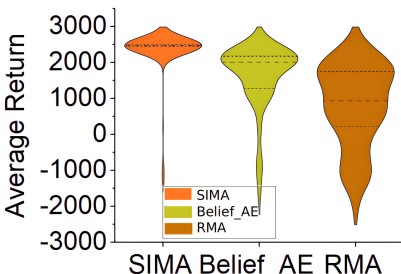

Figure 18: Reward statistics under multiple-sensor noise.

