# OpenReview forum: "Can Agent Learn Robust Locomotion Skills without Modeling Environmental Observation Noise?"
_ICLR.cc/2024/Conference — Submitted to ICLR 2024_

### Official Review · Reviewer_UYTE · 2023-10-29

**Soundness:** 3 good
**Presentation:** 4 excellent
**Contribution:** 3 good
**Rating:** 6
**Confidence:** 4

**Summary:**

This work proposes a self-supervised method, self-supervised randomized masked augmentation, to discover the correlation of multi-variable time series data to reconstruct the optimal state representation. The authors give theoretical guarantees and verify the method's empirical improvement on locomotion tasks for RL algorithms.

**Strengths:**

- The paper proves the effectiveness of the method from both theoretical and experimental perspectives.

- The paper is well written. The authers present the whole framework clearly, and the experimental part explains the advantages of the method with several questions.

- The paper has the case studies to demonstrate the effect of state reconstruction, and the policy distribution unorder the reconstructed states.

**Weaknesses:**

- The paper shows the algorithm boosts the stability of the training process. However, it does not consider whether the method is robust when the test environment has random noises with different distributions. The auothers should consider if the policy is robust in unseen noise distribution, which is much more essential for using RL in the real-world.

- The paper mainly compares whether the training curves of various RL algorithms is better than SIMA when combined with some relatively simple encoders (filters, lstm). However, I think this is slightly unfair for state representation learning. The process of SIMA is obviously more complicated and uses more training objectives. The author should compare it with more solid works on state representation learning for locomotion are, such as [1][2].


[1] Takahiro Miki, Joonho Lee, Jemin Hwangbo, Lorenz Wellhausen, Vladlen Koltun, Marco Hutter "Learning robust perceptive locomotion for quadrupedal robots in the wild"

[2] Ashish Kumar, Zipeng Fu, Deepak Pathak, Jitendra Malik, "RMA: Rapid Motor Adaptation for Legged Robots"

**Questions:**

- Why do we have to use RL algorithms to get a teacher policy? Can we consider using classic control algorithms to generate teacher behaviors?
- Can the SIMA still perform better on a totally unseen noise distribution (after the student policy learning stage)? Is it still robust when noise distribution is shifted?
- In Appendix E, how are the scores calculated? Why can scores above 99 be achieved under clean experimental settings?
- How did the authors draw policy distribution? Is it t-SNE visualization of action? How the actions are sampled?

---

> ### Author Response · Authors · 2023-11-21
>
> Thanks for your positive comments. To provide a clearer answer to your question, the experiment descriptions in section 5.2, 5.3, and Appendix F are revised totally. Furthermore, a new experiment is provided in Appendix J.
>
> **For Weaknesses 1,**
>
> As suggested, we add sufficient descriptions about unseen noise distributions in section 5.3 as below：
>
> Generalization is an important metric for deep learning algorithms. In this experiment, we evaluate if SIMA can be generalized to the environments with various unseen environmental noise. Specifically, we adopt both stationary noise (high-frequency noise) and non-stationary noise (intermittent disturbance), each with 9 groups of different noise characteristics. The stationary noise groups follow gaussian process with frequency mean as $f=[16,32,64]$ (khz), and amplitude variance as $A=[0.4,0.7,1.0]$. The corresponding results are illustrated in Figure 6 (a). Alternatively, the non-stationary noise groups are set to be unpredictable in occurance and duration periods. In such case, we name it as intermittent disturbance which follows uniform distribution with duration period characteristics as $T=[10,15,20]$ (steps), and amplitude characteristics as $A=[5,10,15]$. All these noise durations occur in range of $0$ to $T$ randomly as illustrated in Figure 6 (b). All experimental results are collected from HalfCheetah environments. It can be seen that SIMA outperforms RL-vanilla by $95\%$ in task success rate (we define task success once an agent never falls down during running). Furthermore, SIMA also performs significantly smaller variance in average return than RL-vanilla, which indicates SIMA maintains more stable under various unseen environmental noise.
>
> **For Weaknesses 2,**
>
> As suggested, we add an additional experiment to make comparisons with other well-adopted robotics locomotion state representation learning baselines in Appendix J as below:
>
> In this section, we mainly focus on comparing the performance of SIMA with Belief\_AE [1] and RMA [2].
>
> Figure 15 and Figure 16 show experimental results under single-sensor noise conditions, i.e., during entire locomotion task episodes, observation noise only appears in one signal-input channel of the sensors. Figure 16 shows the average return performance among SIMA, Belief\_AE, and RMA by 50 episodic trials. It can be clearly seen in Figure 16, SIMA and Belief\_AE performs better than RMA.
>
> Figure 17 and Figure 18 show experimental results under multi-sensor noise conditions. Figure 18 shows that SIMA also outperforms Belief\_AE by $25\%$ in average return. More specifically, the "Belief state encoder" in Belief\_AE leverages an attentional gate explicitly controls which aspects of exteroceptive data to pass through. However, the Belief\_AE could handle noises only in one-demensional observaton (e.g., the elevation map in [1]), and lacks of an important mechanism to assess the noise scale for all sensors input.
>
> **For Questions 1,**
>
> Yes, it is feasible to employ classic control algorithms for generating teacher behaviors. However, there will be huge human involving workloads to design different controllers for each environment. On the contrary, RL based teacher policy can learn locomotion skills in various environments, without human involving.
>
> **For Questions 2,**
>
> Please refer to the answer to Weaknesses 2.
>
> **For Questions 3,**
> This part was indeed not explained clearly. All corresponding experiment descriptions are revised totally in Appendix F.
>
> We extensively train individual locomotion control policies in six clean environments (no environmental noise), and record the average returns (e.g., CartPole-200, Halfcheetah-2500). We then adopt these results as benchmarks. To map the experimental results from these different algorithms into a unified comparison range, we collect returns from 30 trials that are all normalized into values between 0.0 and 100.0 as shown in Table 1, in which clean indicates there is no observation noise in the evaluation environments.
>
> All these algorithms achieve above 99 scores in all six clean environments. It indicates all these algorithms perform normally without  noise.
>
> **For Questions 4,**
>
> Yes, we utilize t-SNE to visualize policy distributions.
>
> Firstly, we train a teacher policy with state truth as a reference (illustrated as red dots scattered in Figure 5), a student policy to reconstruct the optimal policy embeddings under these unexpectable disturbance, and a vanilla DRL policy to serve as the control group.
>
> Secondly, these policies are deployed in two evaluation environments, with stationary noise and non-stationary noise, respectively. The teacher policy can directly observe the state truth, while the SIMA policy and the vanilla DRL policy make decisions by the observations with environmental noise.
>
> Consequently, actions of 30 trials are recorded to compare the differences between the distribution of the three policies.

---

> > ### Comment · Reviewer_UYTE · 2023-11-22
> > **Thank you for the comments**
> >
> > Thank the authors for the detailed comments. My questions including *"how to do visaulization"* and *"classical controller as teacher policy"* have been addressed. However, I still think the overall framework is too complex, which makes me expect on a better performance in more realistic and challenging environments. I believe the authors can significantly improve the quality of this paper with more **real-world level** experiments, and I can not raise my score in this stage.

---

### Official Review · Reviewer_6VDo · 2023-11-01

**Soundness:** 2 fair
**Presentation:** 2 fair
**Contribution:** 2 fair
**Rating:** 3
**Confidence:** 4

**Summary:**

The paper proposes a reinforcement learning framework under correlated observational noise. The paper assumes that the ground-truth state observation can be recovered from different sensor modalities and proposes a denoising method by using a masked reconstruction technique. Once the training trajectories are denoised, the paper then trains a student policy using the denoised trajectories through imitation learning. Experiments are done for a set of locomotion tasks built on top of pybullet.

**Strengths:**

The paper deals with an important problem of learning robust policies for sim2real transfer and is well motivated by the multi-sensory integration observation. The approach is overall sound.

**Weaknesses:**

1. The main experiment shows marginal improvements over the baseline. Figure 4 is quite suspicious: SIMA mostly performs the same as the baselines but will have sudden drops in performance, after which the performance will rise again. This suggests a mistake in the experimental setting or visualization.

2. The idea of masked modeling for reinforcement learning is not new. See [1], [2]. Masked modeling itself could help improve the sample efficiency and robustness to noises. The paper does not make comparisons with these methods.

3. The writing of the paper could be further improved, by reducing the use of long abbreviations, e.g. VICOR, RMA (the name is also used in a different sim2real transfer for locomotion paper), MASOR

[1] Masked World Models for Visual Control, Seo et al. CoRL 2022
[2] Multi-View Masked World Models for Visual Robotic Manipulation, ICML 2023

**Questions:**

See weakness.

---

> ### Author Response · Authors · 2023-11-21
>
> Thanks for your review. The experiment descriptions in section 5.1 are revised totally, and an additional comparison with MWM and MV-MWM is conducted in Appendix I.
>
> **For Weaknesses 1,**
>
> This part was indeed not explained clearly and the performance drop is normal.  It's worth noting that SIMA consists of three training stages. In the first two stages, SIMA utilizes clean observations to train a teacher policy and a de-noising module. Due to the fact that the teacher policy of SIMA agent in the first two stages was trained without environmental observation noise, it performs similar to other algorithms in the first half of the training curves in Figure 4.
>
> In the last stage, we initialize a student policy network to complete robust locomotion control tasks in noisy environments, which utilizes optimal state representations learned in the first two stages. Since the student policy lacks of guidance from the teacher policy in the first two stages, a notable performance drop pops-up in the middle section of the learning curves as shown in Figure 4. Consequently, after a short warm-up period intervened by the teacher policy, the performance of SIMA outperforms other algorithms. To give a clearer description of the three stages of SIMA, we provide a schematic diagram in Figure 8 in Appendix E.
>
> **For Weaknesses 2,**
>
> The overall motivation, pipeline, and architecture among Masked World Models (MWM), Multi-View Masked World Models (MV-MWM), and SIMA are totally different. One thing in common is that they all use masks. The differences can be summuarized as three-fold:
>
> (1) Motivation: MWM and MV-MWM mainly focus on sample efficiency enhancement for model-based RL, but SIMA mainly focuses on enhancing system robustness against environmental noise, especially for non-stationary noise with model-free RL.
>
> (2) Pipeline: MWM and MV-MWM employ a 2-tier sampling-training cycles, and iterate untill converge. In contrast, SIMA employs a "never turn back" training mode with 3 stages.
>
> (3) Architecture: MWM and MV-MWM employ an end-to-end MBRL learning architecture, but SIMA proposes a novel teacher-student de-noising policy distillation architecture.
>
> To make these differences descriptions clearer, we add an additional experiment to make further comparations about this in Appendix H. Here, we find an interesting phenomenon in Figure 13. From this figure, it can be clearly seen that SIMA shows significant improvements in learning efficiency compared to MWM in the first two training stages. We speculate the proper reason is the teacher policy training (stage 1) can directly observe the system state truth, thereby makes SIMA a quicker learner in locomotion skills. The corresponding locomotion skills bring correct state and action distributions that are critical to de-noising module learning performance in stage 2 subsequently. In contrast, MWM cannot directly observe system state truth in all training stages. Afterwards, we found that the SIMA algorithm shows a significant performance improvement on average returns after being connected to student policy distillation (stage 3). The essential reason is that SIMA fully learned the optimal state representations under environmental noise conditions by robot running procedures during the first two stages, thus effectively suppress the environmental noise encountered.
>
> To further demonstrate this conclusion, we conduct an additional experiment on the core differences between MWM and SIMA. In case of MWM, the sampling-training cycles of MWM last for many rounds, and in each new round, due to the drastic change of robot running policy distribution. MWM needs to re-adapt to brand new world models and de-noising modules in all training rounds. In view of this, we list the probability distributions of states and actions for multiple MWM training rounds in Figure 13. It can be clearly seen that there have been significant changes in the probability distributions of states and actions corresponding to the new trained robot's running skills in each round, resulting in unstable updates to the MWM policy. In contrast, SIMA employs a "never turn back" training mode throughout the entire training process. Once the first two stages have properly learned de-noising skills, stage 3 only needs to complete single round of student policy distillation based on this. This ensures that SIMA only needs to adapt to new de-noising probability distribution of states and actions once, and thus achieves a better learning performance. This novel learning pipeline of SIMA brings significant improvement in learning curves observed in Figure 14.
>
> Furthermore, Multi-View Masked World Models (MV-MWM) mainly enhances multi-viewport image inputs based on MWM, and the essential difference with SIMA is described as aforementioned.
>
> **For Weaknesses 3,**
>
> We abandon the complex abbreviations, e.g., VICOR, RMA, and MASOR, etc as suggested. Now the overall expression looks clearer and easy to comprehend.

---

> > ### Comment · Reviewer_6VDo · 2023-11-21
> > **Thank you**
> >
> > I appreciate the authors for providing a thorough response to my questions.
> >
> > I acknowledge that SIMA is motivated differently from the masked world model (MWM) and includes an additional teacher-student learning component. The authors provide a comparison with a VAE version of MWM in Fig 13. However, the performance shows marginal performance. Given the marginal performance, the complexity of SIMA becomes a disadvantage, with three-stage teacher-student training. Furthermore, it looks like MWM performance is still increasing. Finally, fig.14 is incomprehensible.
> >
> > Overall, I think the paper is well-motivated and has the potential to become a strong paper. However, the paper needs significant improvement in terms of presentation and experimental comparisons. I cannot recommend acceptance in its current form.

---

> ### Author Response · Authors · 2023-11-22
>
> Thanks for your reply.
>
> As suggested, we add two additional experiments to further prove SIMA's three-stage training pipeline (teacher-student de-noising policy distillation) works significantly better than MWM in terms of learning performance, faster convergency, and training scores.
>
> 1. "...marginal performance...",
>
> * In Figure 13 (a), at $160 \times 10^4$ time steps (point B), due to the intervention of student policy distillation (in stage3), SIMA quickly converges with **almost no variances** (2492.63±31.48). In contrast, MWM still shows significant fluctuations in scores with large variance (1387.55±227.84) . It is still a long way for MWM to converge.
>
> * Furthermore, MWM reaches its highest training score at $220 \times 10^4$ time steps (point C), which needs **40%** time steps more than SIMA. It indicates SIMA has faster convergency.
>
> * We evaluated the learning performance of SIMA (trained for $160 \times 10^4$ time steps) and MWM (trained for $220 \times 10^4$ time steps). As depicted in Figure 13 (b), our approach SIMA (2486.77±**67.42**) exhibits more stable performance and higher traning scores than MWM (2289.43±**276.33**). Large variance indicates that MWM couldn't converge to a stable policy until the end of training.
>
> 2. “MWM performance is still increasing ”,
> * We extended the training time steps to $250 \times 10^4$ and MWM **stops learning** after $220 \times 10^4$ time steps (point C) anymore. Furthermore, there still exists a obvious gap between MWM and SIMA as shown in Figure 13 (b).
>
> 3. "...fig.14 is incomprehensible."
>
> * MWM employs a 2-tier sampling-training cycles, and iterate untill converge. This will result in a significant change in the distribution of sate-action pairs after each update of MWM's policy as shown in Figure 14. Therefore, it needs to readapt to brand new world models and de-noising modules in all training rounds, which also causes bigger variance during training. However, SIMA only needs to adapt once through the entire training process, due to the intervention of student policy distillation. Thus, teacher-student policy traning ensures faster convergency of SIMA and it is necessary for SIMA.
>
> If the concerns have been addressed, we hope the rating could be improved.

---

> > ### Comment · Reviewer_6VDo · 2023-12-02
> > **Final comments**
> >
> > I thank the authors for answering my questions and for coming up with experiments in such a short time. The additional experiments clarify some of my concerns, but I remain skeptical about the improvement of SIMA over MWM.
> > * In convergence, the performance gain seems small compared to the overall score. It is hard to tell how is the score improvement reflected in the learned locomotion policies qualitatively. This is especially a concern given SIMA's more complex training setup.
> > * I think another round of review would be essential where the authors can compare with MWM on all environments.
> >
> > Another main reason for my recommendation of rejection is the current quality of the paper writing and presentation. The main framework of 3-staging training is unclear to me (and also to reviewer GwWq). Both the method section and Figure 4 should be updated and should provide clear motivation on why the three-stage separation is necessary.

---

### Official Review · Reviewer_GwWq · 2023-11-01

**Soundness:** 4 excellent
**Presentation:** 4 excellent
**Contribution:** 3 good
**Rating:** 8
**Confidence:** 4

**Summary:**

For DRL-based locomotor control tasks, the ability to adjust to environmental observation noise is essential. Due to the existence of non-stationary noise, modeling is difficult or impossible. Previous works lack effective solutions for the above problems. In this paper, the authors present a method for learning robust locomotion skills without explicitly modeling noise from environmental observations. Inspired by multi-sensory integration mechanism, the authors first formulate the MDP with an environmental de-noising process as a DRPOP problem. Based on this, the authors propose a Self-supervised randomIzed Masked Argumentation (SIMA) algorithm to learn the internal correlation of multivariate time series and reconstruct latent state representation from noisy observations. The experiments on locomotion control tasks demonstrate that the proposed algorithm performs robust locomotion skills under environmental observation noise, and outperforms state-of-the-art baselines by 15.7% in learning performance.

**Strengths:**

I would like to appreciate the authors for the submission of such high-quality manuscripts.

From a high-level point of perspective, this appears to be a very well-structured work. Related work covers deep reinforcement learning, observation de-noising, and masked multivariate time series modeling completely. The authors are really good storytellers and the topic is novel. This paper is well written and each section is well detailed, especially the introduction section and the methodology section, which is precisely to the point. I truly enjoy every figure. The authors put a great deal of effort into refining the figures.

**Weaknesses:**

1. In section 5.1 LEARNING PERFORMANCE UNDER ENVIRONMENTAL OBSERVATION NOISE (RQ1), the performance of the proposed algorithm shows sharp declines in all locomotion control tasks. I suggest the authors to add more contents to analyze the causes in the manuscript. What's more, more training steps should be visualized since the performance of some baselines is still increasing such as RL-DR and RL-vanilla in the Walker2D environment.

2. There are so many abbreviations throughout the paper, e.g., DRPOP, SIMA, VICOR, RMA, and MASOR, making them difficult to understand. In my humble opinion, one or two abbreviations are good.

**Questions:**

1. I suggest the authors to attempt BetaVAE and Beta-TC-VAE as the generative model to encode masked state observation into the the latent state representation and reconstruct state observation from latent state representation.

---

> ### Author Response · Authors · 2023-11-21
>
> Thanks for your positive comments. The experiment descriptions in section 5.1 are revised totally, and a new experiment about generative models is conducted in Appendix G.
>
> **For Weaknesses 1,**
>
> This part was indeed not explained clearly and performance drop is normal.
>
> It's worth noting that SIMA consists of three training stages. In the first two stages, SIMA utilizes clean observations to train a teacher policy and a de-noising module. Due to the fact that the teacher policy of SIMA agent in the first two stages was trained without environmental observation noise, it performs similar to other algorithms in the first half of the training curves as illustrated in Figure 4.
> In the last stage, we initialize a student policy network to complete robust locomotion control tasks in noisy environments, which utilizes optimal state representations learned in the first two stages. Since the student policy lacks of guidance from the teacher policy in the first two stages, a notable performance drop pops-up in the middle section of the learning curves as shown in Figure 4. Consequently, after a short warm-up period intervened by the teacher policy, the performance of SIMA outperforms other algorithms.
>
> To give a clearer description of the three stages of SIMA, we provide a schematic diagram in Figure 8 in Appendix E. Furthermore, all corresponding experiment descriptions in Section 5.1 are revised totally and carefully to provide a clearer expression.
>
> As suggested, we add more training steps in the Walker2D-v0 ($260 \times 10^{4}$ to $300 \times 10^{4}$ ) and  Halfcheetah-v0 ($200 \times 10^{4}$ to $240 \times 10^{4}$). The results in Figure 4 show that all algorithms have converged. It can be clearly seen that SIMA outperforms other state-of-the-art methods in learning performance.
>
> **For Weaknesses 2,**
>
> We abandon the complex abbreviations, e.g., VICOR, RMA, and MASOR, etc as suggested. Now the overall expression looks clearer and easy to comprehend.
>
>
> **For Questions 1,**
>
> As suggested, we conduct a new experiment to evaluate BetaVAE, and Beta-TC-VAE as the generative models to encode masked state observation in the third paragraph of Appendix G as below:
>
> Generative Models. We attempt BetaVAE, and Beta-TC-VAE as the generative models to encode masked state observation into the the latent state representation. Experimental environment is Halfcheetah-v0 with observation noise. For BetaVAE model, we set three sets of parameters $\beta = 1$ (i.e., vanilla VAE), $\beta = 4$, and $\beta = 50$. For Beta-TC-VAE model, we set the parameters as $\alpha=1$, $\beta = 1$, and $\gamma=6$. The results can be seen in Figure 11. The four generative models have almost the same impact on SIMA performance, with both mean and variance being close. BetaVAE and Beta-TC-VAE aim to balance latent channel capacity and independence constraints with reconstruction accuracy. The locomotion control experimental scenarios in this paper belong to continuous dynamic system. Good disengagement could not result in a significant improvement in downstream continuous control tasks. Experimental results indicate that vanilla VAE can satisfy the requirements of SIMA.

---

### Meta-Review · Area_Chair_uFUm · 2023-12-07

**Metareview:**

*Summary*: This paper aims to design an RL framework with environmental observation noise. First, the actual state conversation is recovered from multi-modal sensors using a denoising method with a masked reconstruction technique. Then, a student policy is trained using the denoised trajectories through IL. Simulation experiments for locomotion tasks show the effectiveness of the proposed framework.

*Strength*: (1) The target problem (RL with correlated observational noise) is critical for many sim-2-real tasks. (2) The proposed framework based on multi-sensory state and recovery is somewhat novel. (3) Writing is mostly clear.

*Weakness*: (1) The framework is proposed with strong motivations from sim2real. However, all experiments are in rather standard and simple Pybullet simulations. The effectiveness in real-world settings is highly uncertain. In locomotion tasks, the higher average return in sim often does not lead to improvement in locomotion skills in real. (2) The proposed framework overall is too complicated.

**Justification For Why Not Higher Score:**

See the weakness part.

**Justification For Why Not Lower Score:**

See the strength part.

---

### Decision · Program_Chairs · 2024-01-16

Reject